# Food Insecurity and Associated Factors among Households in Maputo City

**DOI:** 10.3390/nu15102372

**Published:** 2023-05-18

**Authors:** Elias M. A. Militao, Olalekan A. Uthman, Elsa M. Salvador, Stig Vinberg, Gloria Macassa

**Affiliations:** 1Department of Health Sciences, Faculty of Humanities, Mid Sweden University, Holmgatan 10, 851 70 Sundsvall, Sweden; stig.vinberg@miun.se; 2Department of Public Health and Sports Science, Faculty of Occupational and Health Sciences, University of Gävle, Kungsbacksvägen 47, 801 76 Gävle, Sweden; 3Department of Biological Sciences, Faculty of Science, Eduardo Mondlane University, 3453 Julius Nyerere Avenue, Maputo 257, Mozambique; elsamariasalvador@gmail.com; 4Warwick Centre for Global Health, Division of Health Sciences, Warwick Medical School, University of Warwick, Coventry CV4 7AL, UK; olalekan.uthman@warwick.ac.uk; 5Department of Global Health, Division of Epidemiology and Biostatistics, Faculty of Health Sciences, Stellenbosch University, Francie van Zijl Drive, Cape Town 7505, South Africa; 6EPI Unit—Instituto de Saúde Pública, Universidade do Porto, Rua das Taipas 135, 4050-600 Porto, Portugal

**Keywords:** associated factors, food insecurity, prevalence, southern Mozambique

## Abstract

Food insecurity (FI) is a global concern and is one of the main causes of malnutrition in low- and middle-income countries. In Mozambique, the burden of FI and how various factors contribute to FI is not well known. This study aimed to investigate the prevalence of FI and its associated factors in southern Mozambique. Data from 1842 household heads in Maputo City were analyzed in a cross-sectional design. Food insecurity was measured using a modified version of the US Department of Agriculture Household Food Security scale, and its association with socio-demographic factors was assessed through multiple regressions. Altogether, 79% of the households were food insecure; of these, 16.6% had mild FI, 28.1% moderate and 34.4% severe FI. The study revealed that low-income households, those with less educated heads, and those engaged in informal work were significantly more prone to FI. Likewise, dietary diversity and the number of meals were also significant predictors of FI. These findings suggest the need for decent work and job creation, which calls for joint efforts from government, the private sector, and international institutions. Furthermore, these key drivers should be considered in the development of public health policies and programs designed to alleviate household FI and malnutrition in Mozambique.

## 1. Introduction

Food insecurity (FI) is a global concern and a visible reality for most people in low- and middle-income countries (LMICs) [1], and has become an urgent public health issue that affects nourishment, health, and human well-being worldwide [2,3]. Food insecurity is defined as “a limited or uncertain availability of nutritionally adequate or safe foods or limited or uncertain ability to acquire food in socially acceptable ways” [4,5]. Others describe FI as a set of situations in which households compromise on the quality and variety or quantity of food as a consequence of deficient household resources and/or inability to afford food [3].

According to a report by FAO et al. [1] (pp. 14–15), about 768 million people worldwide (9.8% of the world’s total population) are undernourished. Of these, 760 million are in developing countries, of whom 278 million live on the African continent and 261 million in Sub-Saharan Africa (SSA). Within SSA, Southern Africa emerges with a more favorable scenario while the Central appears with the worst picture, though the highest burden in numbers remains in the Eastern Africa region [1]. In addition, estimates indicate that about 2.3 billion people (29.3% of the global population) are moderately or severely food insecure. The number of undernourished people worldwide has increased, highlighting the challenge of achieving the United Nations’ Sustainable Development Goal (SDG) of Zero Hunger, eradicating FI and all forms of malnutrition by 2030. In fact, projections indicate that nearly 670 million people will still face hunger in 2030, and currently, the number of people unable to afford a healthy diet has risen dramatically, especially because of conflicts, climate change and the COVID-19 pandemic [1]. The Russia–Ukraine war, the COVID-19 pandemic, along with climate change reinforce the need to rethink future actions towards global food security (FS) and nutrition, especially concerning staple foods [6,7]. Research evidence suggests that household FS and nutrition in many LMICs is much worse due to these factors combined with deficient socioeconomic and political conditions [1,6,8]. To this end, government policies along with agricultural investments (e.g., tax and tariff reductions, fertilizer subsidies, and rural infrastructure) are critical factors to protect domestic consumers and producers from rise in reference (border) price for imported food and agricultural inputs while ensuring that policy actions and investments needed for long-term agricultural development are not jeopardized, especially in African countries [9,10,11]. Likewise, rapid population growth is a serious challenge that needs to be considered when addressing global food security and nutrition [12,13].

In Mozambique, FI continues to be the primary challenge to economic growth and human well-being [14]. According to FAO [15], Mozambique is a low-income, high-inequality and food-deficit country with a largely rural population of 28 million. It ranked 181st out of 189 countries in the 2020 Human Development Index, 106th out of 116 in the 2021 Global Hunger Index and 127th out of 162 in the 2019 Gender Inequality Index [16]. The country achieved some progress under Millennium Development Goals target 1C of halving the number of undernourished people, but significant challenges remain that will hinder progress towards the targets of SDGs 2 [17]. About 80% of the Mozambican population cannot afford an adequate diet [18] and almost 60% live in extreme poverty [19]. Most households are at risk of FI and of adopting negative coping strategies [17,20]. Furthermore, Mozambique is prone to regular droughts, and floods occur virtually every year in large watersheds and poorly drained urban settlements. Roughly 70% of the rural population depends on subsistence agriculture, the markets are not integrated and post-harvest losses reach 30% due to limitations in storage, processing and handling [17,21]. However, the extent and severity of FI in general, as well as its associated factors, are not well known. Some reports suggest poverty, food unavailability, inadequate food intake and a non-diversified diet, high levels of infectious disease, poor knowledge of healthy foods, inadequate food preparation and eating habits as the underlying causes of undernourishment among the population [14,22]. A study by McCordic et al. [23] found female headship to be strongly linked to household FI, even though the degree of this relationship was somewhat dependent on employment and education. This study sought to investigate the prevalence of and factors associated with household FI in Maputo City, southern Mozambique.

## 2. Materials and Methods

### 2.1. Study Setting and Sampling

A cross-sectional study was conducted in Maputo, southern Mozambique. Maputo is the capital city of Mozambique and is divided into seven municipal districts that include KaNyaka Island and Katembe across the bay [24]. Maputo is the largest urban agglomeration in Mozambique. The city has a high level of underemployment, with most people being engaged in informal work [25]. Its urban landscape is usually divided into three areas. The first, the KaMpfumu district, is the wealthiest area of the city. The second consists of the poorer residential suburbs and covers Nlhamankulu and KaMaxaquene. The third covers the peri-urban districts of KaMavota and KaMubukwana [24]. According to the 2017 general census, Maputo has about 1,080,277 inhabitants (52% female and 48% male) and 235,750 households [26]. About 71% of the households are food insecure [23].

The selection of households included in this study relied on a two-stage design inspired by the National Institute of Statistics platform which was used by the Technical Secretariat for Food Security and Nutrition (SETSAN) in their 2013 Baseline Study [27]. Therefore, enumeration areas (located in a suburb or peri-urban district of Maputo) were randomly selected in the first stage. A total of 96 enumeration areas were selected, and each enumeration area provided a maximum of 20 households. Next, households within each enumeration area were selected using a systematic random sampling strategy (to sample every fifteenth household). Households whose heads (or other representative household members) did not meet the age criterion (18–60 years), or that did not agree to participate or decided to drop out were excluded from the study. In collaboration with the municipality, eligible households were approached and informed about the objectives and voluntary nature of the study. They were invited to participate and had up to 5 days to respond to the invitation letter. The total sample size was 1842 households based on approximate proportional allocation.

### 2.2. Data Collection and Procedure

Face-to-face structured interviews were performed at each participant’s home between November 2021 and June 2022. Each interview lasted about 45–60 min. Data from households were collected using a questionnaire previously validated in Portuguese speaking populations [28,29,30] but adapted to the Mozambican context. The questionnaire consisted of eight items from the United States Department of Agriculture Household Food Security Survey Module (USDA HFSSM) designed to measure FI in the last 3 months. In addition, the questionnaire included background demographic and socioeconomic questions (education, work, income) as well as questions about: (a) physical health (diagnosed hypertension and type 2 diabetes); (b) mental health/psychiatric disease; (c) medication and health care utilization; (d) self-reported health; (e) food consumption patterns (dietary items, from oils, vegetables, meat, fish, starch to beverage); (f) barriers to food access; (g) food expenses and purchasing habits; (h) own food production or other ways to obtain food; (i) sleeping patterns; (j) life style/physical activity; and (k) health behavior (smoking and alcohol consumption). Before data collection, the questionnaire was piloted by the authors E.M. and E.S. in Manhiça district (a region outside the study setting) to ensure that it was accurate and effective.

The original, 14-item USDA HFSSM was modified to focus only on adult and household FI by excluding the six items related to children (Table 1). To make the scale easier to use, a “yes/no” response format was adopted. All “yes responses” were followed up by asking, “How often did it occur?”, with three response options (often, sometimes, rarely). The responses “often” and “sometimes” were coded as 1, while “rarely” was coded as 0. Consequently, the scale had a maximum of 8 points. Households were considered food secure if they scored 0 or 1 and food insecure if they scored ≥2. Among food-insecure households, those that scored 2 or 3 were considered to have mild FI, while those scoring 4–6 were classed as having moderate FI, and scores of 7 or 8 were considered to represent severe FI. Similarly, regarding dietary patterns, three categories were considered (low, medium and high dietary diversity) based on food items consumed in the last 7 days [31]. Low dietary diversity included diets consisting largely of cereals, tubers and roots, vegetables and greens, oils, and fish and other seafood. The medium category added meat or poultry, spices, and eggs to the list. The last category, high dietary diversity, included diets composed of cereals, vegetables and greens, oils, spices and condiments, fish and other seafood, meat or poultry, eggs, fruits, beverages, and dairy products. With regard to occupation, the term “unpaid work” was used to categorize household heads engaged in informal work (insecure employment mostly self-employment), while “paid work” was used to categorize those engaged in formal work (secure employment mostly from the government and the private sector). Finally, both food transfers and money transfers from relatives (largely from South Africa) were considered to constitute remittance in this study.

### 2.3. Data Analysis

All data analyses were performed in IBM SPSS Statistics 27 [32]. Descriptive statistics was carried out and frequencies and percentages were used for categorical variables, and means for continuous variables. Before multiple regression was performed, preliminary analyses were conducted to ensure that the assumptions of normality, linearity, multicollinearity and homoscedasticity had not been violated. Multiple regression was performed to explore the predictive ability of a set of variables to explain the outcome variable (FI score), as well as to assess the relative contribution of each individual variable. Standard multiple regression was then performed to gain a quick insight into which predictors are relevant for explaining FI. Lastly, stepwise multiple regression was employed with all predictors as performed in standard multiple regression to check the consistency between the two techniques. In addition, as households were categorized as food secure versus food insecure, binary logistic regression was performed to explore the predictive power of the same set of variables (used on multiple regression) to explain the outcome variable (FI). A 95% confidence interval and *p*-value ≤0.05 were used to assess the statistical significance of the association between the explanatory variables and the outcome variable.

### 2.4. Ethical Approval

The study protocol was approved by the Institutional Committee of Bioethics in Health of the Faculty of Medicine, Eduardo Mondlane University, Maputo (registration No. CIBS FM&HCM/036/2019). Informed consent was obtained from each participant prior to the data collection, and all ethical requirements (e.g., voluntariness, confidentiality, anonymity) were followed.

## 3. Results

### 3.1. Characteristics of the Study Participants

This study included 1842 households located in Maputo City, representing a participation rate of about 97%. Table 2 illustrates the socio-demographic characteristics of the study population. Most households were male headed (71.6%), had five members, and 66.6% were married or living in marital union. In addition, 45.4% of the heads of households had primary or secondary education, and 48.5% did not have secure employment and were in the informal sector.

### 3.2. Prevalence of Household Food Insecurity

In this study, Cronbach’s alpha was 0.871. Of the 1842 households that participated in the study, only 21% were food secure, while 79% were food insecure. Among food-insecure households, 16.6% were experiencing mild FI, while 28.1% and 34.4% were experiencing moderate and severe FI, respectively.

In the regression analyses, FI was associated with various factors. The final model was able to explain 75% of the variability (adjusted R Square = 0.752) in FI. Among ten variables included in the models, eight reached statistical significance with focus on household income, dietary diversity, education level, type of work and number of meals per day (Table 3). Likewise, the binary logistic regression revealed that household income, dietary diversity, education level, type of work and number of meals per day were the main predictors of FI (Table 4). The findings showed that households with a monthly income of <9000 Mozambican metical (equivalent to USD 140.86) were ten times more likely to be food insecure than their counterparts who had a monthly income of >15,000 metical (equivalent to USD 234.77). The households whose heads had primary or secondary education, for example, were twelve times more exposed to FI compared with those with university education. Likewise, households whose heads worked in the informal sector were three times more exposed to FI than their counterparts who had secure employment in the formal sector. On the other hand, households whose meals in the past 7 days were of medium diversity were eight times more likely to be food insecure than their counterparts whose meals were of high diversity. Similarly, households eating two meals a day were four times more likely to be food insecure compared with those eating three meals a day.

## 4. Discussion

This study sought to investigate the prevalence of and factors associated with household FI in Maputo City, southern Mozambique. About 62.5% of the study population were experiencing moderate and severe FI, highlighting the challenges that Mozambique faces regarding food security and nutrition. This proportion may be conservative considering that about 60% of the Mozambican population live in extreme poverty [19], and the effects of the COVID-19 pandemic [20,33], as the data collection was carried out just after the pandemic. In any case, moderate and severe FI are a matter of concern for most households not only in Mozambique (and Maputo city in particular) as highlighted in this study, but also across SSA and other LMICs [1,34].

In fact, the SDGs, especially those aiming to end extreme poverty and eradicate hunger and all forms of malnutrition by 2030, are out of reach for many LMICs, including Mozambique [1,35]. This being a decade of informed actions, there is an urgent need to implement both emergency solutions, especially for the most vulnerable groups, and more sustainable solutions to combat household FI and malnutrition, and eventually achieve the SDGs [34,36]. In addition, rapid population growth is a serious challenge that needs to be urgently considered when addressing global food security and nutrition, as overpopulation is expected to be the leading cause of FI worldwide by 2050 [13,37].

In this study, various variables were linked to FI, with a focus on socioeconomic and demographic factors (household income, dietary diversity, education level, type of work and number of meals). Food insecurity in Mozambique, and in Africa in general, is largely linked to poverty [38,39]. Therefore, there is a need to tackle those factors that contribute significantly to poverty and a rise in poverty. Research evidence in Africa indicates that flawed economic policies, corruption, poor governance and political conflicts [40,41] as well as poor land utilization are the main causes of poverty that need to be addressed [40]. Similarly, poor governance and political conflicts [42,43] have been highlighted as relevant determinants of FI in Africa. For instance, the quality of governance (e.g., government effectiveness, rule of law, accountability) and targeted policies are critical to promoting a stable environment that is suitable for economic investments, especially those related to improving food security and nutrition, social protection and the pace of economic growth [43]. To this end, effective interventions from major players such as governments, the private sector and international institutions are essential to stimulate decent work and build an inclusive economic growth for Africa as a whole [44].

Household income and food prices are critical determinants of FI in African cities, as most urban households buy most of their food rather than producing it themselves [38]. Urban FI tends to worsen with high food prices observed in cities. For instance, the cost of living and basic needs in Mozambique, especially in major urban settlements, has been worsening at a great pace [45]. Moreover, the COVID-19 pandemic has caused a rise in FI linked to food shortages and high food prices, job losses and a decrease in livelihoods [20]. A study by Rosenberg et al. [46] in 16 southern African countries found an association between the COVID-19 pandemic and increased risk of job loss. Therefore, the pandemic, in a unique way, has highlighted the weaknesses of food systems, and also of health systems in LMICs [47] as well as in high-income countries [48]. Altogether, the above-cited studies highlight the need to provide social and economic support to the most vulnerable groups [49,50] as well as to rethink future actions towards global food security and nutrition [48,49]. Furthermore, FI in urban settlements is largely a result of low income, often accompanied by a lack of secure employment as highlighted in this study, and by poor living conditions and limited access to clean water, sanitation and electricity [51]. In contrast to the African cities, seasonality and climate change can have a huge impact on household FI in rural areas. Most rural households depend on their own food production, which is heavily dependent on rainfall, which is becoming increasingly unreliable because of climate change [52]. Likewise, climate change may affect food systems and FI in various ways, from its direct effects on food production to changes in markets, food prices and food supply chain infrastructure [53].

On the other hand, various studies have shown a relationship between household FI and gender inequality in Africa and other LMICs, and the need for socioeconomic interventions to empower women and close the gender gap between female- and male-headed households [54,55]. Just as in the study by McCordic et al. [23], FI in the present study was associated with female headship depending mostly on their education level and employment. Similar results have been found across the world [54,56]. Indeed, gender inequality has been globally recognized as a significant determinant of household FI [56,57,58]. Due to inequalities in education and employment across female- and male-headed households, female-headed households are more likely to suffer from FI [23,57]. Likewise, there is inequality in social safety nets available to women [56,59]. Additionally, factors such as cultural norms and values, ownership of quality land and productive resources, including provision of extension services, compound the household FS situation in rural areas, as these factors tend to favor men over women [59]. Moreover, households with five or more members (including several adult members) were more likely to be food secure than their counterparts who had few members, which suggests that household size can have a protective effect against FI. Considering that most households were composed of members with considerable financial constraints, the dependency ratio could be more pronounced in smaller households than in larger ones. Indeed, a large household with more productive adults contributing to the affairs of that household exhibits improved household FS status [57,60]. In addition, there is substantial evidence showing the link between FI and dependency ratio [36,44]. Similarly, as highlighted in this study, research evidence across the world indicates that single parents, especially single mothers, are more likely to be food insecure than their counterparts who are married or in marital union [61,62,63].

At the same time, besides food quality and safety, food quantity and dietary diversity are a real concern for the vast majority of poor urban households [20,31]. In fact, households experiencing moderate and severe FI in this study were strongly associated with low dietary diversity and having very few meals a day. In any case, it is necessary to consider several factors together to get an overview of household FI, as one factor may not fully capture and explain the extent and severity of this complex phenomenon. In summation, the urgency of household FI and malnutrition in Mozambique should be recognized as a call for the country to develop multifaceted and multisectoral programs that include household FS and nutrition as an integral component to improve physical and mental health, especially among the most vulnerable groups. Likewise, these findings reinforce the need for social protection for the most vulnerable groups; in addition, they call for implementation of multifaceted programs to enhance food security and nutrition at a national, regional and global level [20,43,56].

### Strengths and Limitations

This is one of the few quantitative studies that provide empirical and updated information on household FI and its associated factors in Maputo City in the context of the COVID-19 pandemic. In addition, a considerable number of household heads participated in the study, meaning that the findings can be generalized to the entire city of Maputo. Nonetheless, the study has some limitations to be considered. Besides using a cross-sectional design, the respondents’ recall bias needs to be acknowledged, and the findings may not be fully applicable to other Mozambican cities whose characteristics (e.g., socio-cultural, demographic, and economic) are very different from the capital city.

## 5. Conclusions

This study aimed to investigate the prevalence of household FI and its associated factors in Maputo City. Almost four in every five households were food insecure, three of which were experiencing moderate or severe FI. Regarding associated factors, several variables were linked to household FI with a focus on poverty, low income, lack of secure employment, low education level, household size and dependency ratio, non-diversified diet and a very limited number of meals per day. This highlights the dynamic and complex nature of household FI, and the need for longitudinal studies (e.g., case–control studies, participatory action research and ethnographic studies) to gain a deeper understanding of the mechanisms linking demographic and socioeconomic factors to household FI and malnutrition, and to ascertain causality. On the other hand, these findings demand urgent informed actions from the government, the private sector and international institutions, not only to provide social and economic support to the most vulnerable groups, but also to implement multifaceted programs to combat household FI and malnutrition and create more job opportunities and promote decent work.

## Figures and Tables

**Table 1 nutrients-15-02372-t001:** Adapted United States Department of Agriculture Household Food Security Survey Module (USDA HFSSM).

In the last three months:
Were you worried that your food would run out before you had money or other resources to buy more?
2. Did you run out of food before you had enough money or other resources to buy more?
3. Did you have to eat the same foods daily because you did not have money or other resources to buy other foods?
4. Did you or any other adult in your household cut the size of your/their meals because you did not have enough money or other resources to buy food?
5. Did you or any other adult in your household skip some of your/their daily meals because you did not have enough money or other resources for food?
6. Did you ever eat less than you felt you should, because you did not have enough money or other resources to buy food?
7. Were you ever hungry and did not eat because you did not have money or other resources to buy enough food?
8. Did you or any other adult in your household ever not eat for a whole day because you did not have enough money or other resources to buy food?

**Table 2 nutrients-15-02372-t002:** Descriptive characteristics of the study participants, Maputo City, southern Mozambique.

Variables	Maputo Districts Covered by the Study				
	KaMaxaquene (*n* = 474)	KaMavota (*n* = 450)	KaMubukwana (*n* = 467)	Nlhamankulu (*n* = 451)	Total (*n* = 1842)
Head of household					
Male	331 (69.8%)	315 (70%)	316 (67.7%)	356 (78.9%)	1318 (71.6%)
Female	143 (30.2%)	135 (30%)	151 (32.3%)	95 (21.1)	524 (28.4)
Age of participant (mean ± SD)	34.57 ± 9.14	30.91 ± 8.53	32.58 ± 9.61	34.13 ± 8.79	33.05 ± 9.04
Household size (mean ± SD)	4.91 ± 0.9	4.66 ± 0.95	4.74 ± 0.96	4.85 ± 0.91	4.82 ± 0.75
Number of children (mean ± SD)	2.7 ± 0.77	2.41 ± 0.81	2.52 ± 0.86	2.43 ± 0.81	2.54 ± 0.74
Marital status					
Single/separated/divorced	188 (39.7%)	147 (32.7%)	165 (35.3%)	116 (25.7%)	616 (33.4%)
Married/marital union	286 (60.3%)	303 (67.3%)	302 (64.7%)	335 (74.3%)	1226 (66.6%)
Education					
Primary and secondary	200 (42.2%)	202 (44.9%)	221 (47.3%)	214 (47.5%)	837 (45.4%)
High school (Grade 11–12)	210 (44.3%)	182 (40.4%)	202 (43.3%)	168 (37.3%)	762 (41.4%)
University	64 (13.5%)	66 (14.7%)	44 (9.4%)	69 (15.3%)	243 (13.2%)
Type of work					
Unpaid work	233 (49.2%)	211 (46.9%)	229 (49%)	220 (48.8%)	893 (48.5%)
Paid work	241 (50.8%)	239 (53.1%)	238 (51%)	231 (51.2%)	949 (51.5%)
Monthly household income (in MZN)					
<9000	189 (39.9%)	189 (42%)	180 (38.5%)	175 (38.8%)	733 (39.8%)
9000–15,000	160 (33.8%)	139 (30.9%)	184 (39.4%)	151 (33.5%)	634 (34.4%)
>15,000	125 (26.4%)	122 (27.1%)	103 (22.1%)	125 (27.7%)	475 (25.8%)
Remittance from relatives					
No	357 (75.3%)	331 (73.6%)	333 (71.3%)	339 (75.2%)	1360 (73.8%)
Yes	117 (24.7%)	119 (26.4%)	134 (28.7%)	112 (24.8%)	482 (26.2%)
Dietary diversity					
Low	251 (53%)	225 (50%)	277 (59.3%)	222 (49.2%)	975 (52.9%)
Medium	128 (27%)	143 (31.8%)	143 (30.6%)	149 (33%)	563 (30.6%)
High	95 (20%)	82 (18.2%)	47 (10.1%)	80 (17.7%)	304 (16.5%)
Number of meals (mean ± SD)	2.2 ± 0.44	2.25 ± 0.43	2.24 ± 0.41	2.23 ± 0.42	2.23 ± 0.42

Mozambican metical (MZN); standard deviation (SD).

**Table 3 nutrients-15-02372-t003:** Prevalence of household food insecurity by socio-demographic characteristics, Maputo City Household Survey, 2022.

Variables	Prevalence of Food Insecurity (%)		*p*-Value
	Food Insecure (%)	Food Secure (%)	
Head of household			0.001
Female	85.5	14.5	
Male	76.4	23.6	
Household size			0.001
1–4 members	84.8	15.2	
5 or more	75.2	24.8	
Number of children			0.386
0–2 children	77.3	22.7	
3 or more	79.7	20.3	
Marital status			0.001
Single/separated/divorced	87.2	12.8	
Married/marital union	74.9	25.1	
Education			<0.001
Primary and secondary	94.3	5.7	
High school (Grade 11–12)	77.8	22.2	
University	16	84	
Type of work			<0.001
Unpaid work	93.1	6.9	
Paid work	65.3	34.7	
Monthly household income (in MZN)			<0.001
<9000	95.95	4.05	
9000–15,000	86.1	13.9	
>15,000	37.1	62.9	
Remittance from relatives			0.516
Yes	82.6	17.4	
No	77.7	22.3	
Dietary diversity			<0.001
Low	97.85	2.15	
Medium	73.2	26.8	
High	23	77	
Number of meals			<0.001
1–2 meals	93.3	6.7	
3 or more	31.1	68.9	

Mozambican metical (MZN).

**Table 4 nutrients-15-02372-t004:** Binary logistic regression analysis of food insecurity by socio-demographic characteristics, Maputo City Household Survey, 2022.

Variables	OR	95% CI	*p*-Value
Head of household		1.38–2.39	0.001
Female	1.82		
Male	1		
Household size		1.14–2.31	0.007
1–4 members	1.63		
5 or more	1		
Number of children		0.53–1.14	0.196
0–2 children	0.79		
3 or more	1		
Marital Status		1.74–2.98	0.001
Single/separated/divorced	2.28		
Married/marital union	1		
Education			<0.001
Primary and secondary	11.87	4.92–28.62	
High school (Grade 11–12)	4.17	2.19–7.91	
University	1		
Type of work		2.32–4.14	<0.001
Unpaid work	3.1		
Paid work	1		
Monthly household income (in MZN)			<0.001
<9000	10.54	7.84–14.01	
9000–15,000	2.04	1.32–3.15	
>15,000	1		
Remittance from relatives		0.66–1.346	0.747
Yes	0.943		
No	1		
Dietary diversity			<0.001
Low	–	–	
Medium	8.1	5.82–11.23	
High	1		
Number of meals		2.84–5.09	<0.001
1–2 meals	3.8		
3 or more	1		

Odds ratio (OR); confidence interval (CI); Mozambican metical (MZN).

## Data Availability

The data presented in this paper are not publicly available owing to restrictions in the ethical approval for this study. Questions related to the data should be directed to the corresponding author.

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
