# Peer review of "Food Insecurity and Associated Factors among Households in Maputo City"

_nutrients, 2023, doi:10.3390/nu15102372_

Round 1

Reviewer 1 Report

The manuscript titled “Food Insecurity and Associated Factors among Households in Maputo City” deals with an interesting problem that fits perfectly in the journal’s scope. There are almost only minor issues to deal with.

Among the major issues, first, it has to be mentioned that the area covered by the study is limited (you acknowledged that among the limitations); it would have been better if you collected data from various parts of Mozambique. Regarding the sampling method, the sampling plan is missing (especially, how many households you planned to include in the sample based on what criterion – the participation rate refers to the final sample size compared to your planned sample size?). The specific random selection method of areas is missing (as well as the total number of areas), moreover, it is not clear what the relation is between enumeration areas and districts – the formers are part of the latter? Based on Table 2, it seems you chose 4 districts out of 5 or 7, I cannot decide because in line 89 you mention there are 7, then in lines 92-95, you list only 5. You should provide more information about the systematic random sampling strategy as well. Concerning the questionnaire, several items of socio-demographic variables you list in lines 118-128 cannot be found among the results (see groups a, b, c, d, f, g, h, I, j, k); excluding them has to be explained. In turn, several variables among the results are not listed there, see the gender of household head, household size, number of children, marital status, number of meals, and remittance (although, the last one is mentioned in the next paragraph). In the literature review and in the discussion I recommend you to refer to more international studies examining FI, and to compare and contrast your results with the previous findings, focusing on your revealed influencing factors. The results are not surprising, they are rather obvious; therefore, it would have been useful to incorporate the omitted variables in the model that would have been resulted in more novel results (unless you have a specific reason for omitting them).

Minor issues include that in the abstract, recommendations should be more related to the results (i.e., how could public health policies help alleviate FI?). In the Introduction section, “some progress” under MDGs should be specified. In section 2.3 standard deviation should also be mentioned and preliminary analyses results as well. The term “with focus on” seems to be inappropriate in lines 192 and 303, I think you mean these variables showed the lowest p values; however, 3 additional variables were also significant (at p<0.01), so I cannot see why these should not be mentioned. In the Discussion section, I recommend you to change the order of the first two paragraphs, this way it seems to be more logical to me.

Author Response

Dear Reviewer,

Thanks very much for your constructive feedback. Please find attached the responses for your suggestions.

Reviewer 2 Report

Scientifically speaking, this article clearly establishes the objectives of the research on which it is based. It describes in detail the random sampling method used to collect the quantitative and qualitative information from which it starts. It detects the basic problems behind the high level of food insecurity that the work itself reveals and outlines the main lines of action that should lead to future improvement of the worrying data it provides. I believe that the work is overzealous which, at times, is unnecessary, as when it concludes that "households eating two meals a day were four times more likely to be food-insecure compared with those eating three meals a day".

Author Response

Dear Reviewer,

Thanks very much for your constructive feedback. Please find attached the responses to your suggestions.
